# Effect of Substrate Temperature on Morphological, Structural, and Optical Properties of Doped Layer on SiO_2_-on-Silicon and Si_3_N_4_-on-Silicon Substrate

**DOI:** 10.3390/nano12060919

**Published:** 2022-03-10

**Authors:** Suraya Ahmad Kamil, Gin Jose

**Affiliations:** 1Faculty of Applied Sciences, Universiti Teknologi MARA, Shah Alam 40450, Selangor, Malaysia; 2School of Chemical and Process Engineering, University of Leeds, Leeds LS2 9JT, UK; G.Jose@leeds.ac.uk

**Keywords:** Er^3+^-doped glass, laser ablation, optical materials, ultrafast lasers, thin film

## Abstract

A high concentration of Er^3+^ without clustering issues is essential in an Er-doped waveguide amplifier as it is needed to produce a high gain and low noise signal. Ultrafast laser plasma doping is a technique that facilitates the blending of femtosecond laser-produced plasma from an Er-doped TeO_2_ glass with a substrate to form a high Er^3+^ concentration layer. The influence of substrate temperature on the morphological, structural, and optical properties was studied and reported in this paper. Analysis of the doped substrates using scanning electron microscopy (SEM) confirmed that temperatures up to approximately 400 °C are insufficient for the incoming plasma plume to modify the strong covalent bonds of silica (SiO_2_), and the doping process could not take place. The higher temperature used caused the materials from Er-doped tellurite glass to diffuse deeper (except Te with smaller concentration) into silica, which created a thicker film. SEM images showed that Er-doped tellurite glass was successfully diffused in the Si_3_N_4_. However, the doping was not as homogeneous as in silica.

## 1. Introduction

Er-doped fibre amplifier (EDFA) was invented in 1987, and it has now been established as one of the standard components in telecom networks [1]. It has facilitated worldwide information exchange because of its unique characteristics, including low noise [2], high gain [3], low loss [4,5] and low-dispersion wavelengths, especially at C-band (1525–1565 nm) and L-band (1565–1610 nm) of fibre optical communication [6]. In addition, EDFA is an excellent candidate for signal amplification at various points in such networks because it is compatible with fibre light-wave systems [7].

Although EDFA is extensively used in various applications, EDFA integration with other optical and electronic components on a compact integrated platform is complex because of its large size, which causes the packaging to be expensive and a barrier to the downsizing of the device [8,9]. The Er-doped waveguide amplifier (EDWA) concept was introduced to mitigate this issue. Its operation principle is similar to an EDFA but a miniaturised planar version to meet the emerging demands [10]. EDWA inherits EDFA’s magnificent performance but with a smaller size, and it can be fabricated on silicon platforms compatible with complementary metal-oxide semiconductor (CMOS) processing [11].

The selection of the material host for EDWA is crucial as it affects the gain per unit length, which is closely related to the solubility of the dopant Er and its photoluminescence lifetime [12]. Silica-based waveguides have the potential to provide the best characteristics of a host to Er in EDWAs because of their refractive index matching with well-established silica optical fibre that potentially minimises coupling loss [13]. Apart from silica, silicon nitride (Si_3_N_4_) is seen as a potential host material for EDWA applications. Si_3_N_4_ is an attractive candidate because of its high refractive index (1.99–2.29) [14] and because it is a Si-based material, such as SiO_2,_ which is compatible with CMOS processing.

Ultrafast laser plasma doping (ULPD) is a technique that has been proven successful for doped Er^3+^-ions in relatively high concentrations without the occurrence of severe clustering issues in silica-based film, and the obtained layer is referred to as Er-doped tellurite-modified silica (EDTS) [13,15,16]. In this approach, the target glass is ablated using a femtosecond laser. The interfacial reaction between a high energy plasma plume and the heated substrate results in a thin layer consisting of a mixture of target glass and substrate material. This laser-induced plasma-assisted process is different from the deposition of a film on the substrate as in well-known pulsed laser deposition. The surface implantation and dissolution of ions and nanoparticles produced by the femtosecond laser-induced plasma into the substrate surface network results in the structural modification of the substrate surface [17]. Previous studies have proven Er-TZN to be successfully doped into silica [13,15,16,18,19,20]. Since this process is also assisted by substrate heating during the doping process, selecting the appropriate temperature is very critical. The substrate temperature is a significant parameter in ULPD because it activates the mobility of the ablated species for it to diffuse into the substrate and modify the host SiO_2_ and Si_3_N_4_ networks. Strong covalent bonds can be modified by the incoming plasma plume when appropriate temperatures are used to heat the substrate. Therefore, the effect of substrate temperature for these two substrates is worth studying as this effect in the ULPD technique has not yet been reported. The use of different substrate temperatures, thus, results in varying doped layer characteristics.

In this work, the doping of Er^3+^-doped tellurite-based glass in silica-on-silicon (SOS) and Si_3_N_4_-on-silicon substrate using the ULPD technique was presented. The SOS substrate was heated at 400 °C, 570 °C, and 700 °C and the Si_3_N_4_-on-silicon substrate was heated at 400 °C, 520 °C, 570 °C, 600 °C, and 650 °C to study the doping process. The effect of the substrate temperature on morphological, structural, and optical properties of the doped layer on SOS and Si_3_N_4_-on-silicon substrates was studied using the produced samples.

## 2. Materials and Methods

### 2.1. Sample Fabrication

The doped layer on SOS and Si_3_N_4_-on-silicon substrates was fabricated via the ULPD technique [18,21]. The initial thickness for SiO_2_ and Si_3_N_4_ layers on silicon was 1 µm. A commercial Coherent Ti: Sapphire LIBRA laser with a pulse duration of 100 fs, a wavelength centred at 800 nm, a repetition rate of 1 kHz, and an energy of 50 µJ was used in this work. The fs laser beam was focused on the target glass (79.5%TeO_2_–10%Na_2_O–10%ZnO–0.5%Er_2_O_3_, Er-TZN) surface at an incident angle of 60° from normal. The substrate was heated to a specific temperature (400–700 °C) at a ramp rate of 50 °C/min under an oxygen ambient pressure of 70 mTorr. The laser ablation process was carried out for 4 h.

### 2.2. Sample Characterisation

Hitachi SU8230 scanning electron microscopy (SEM, Hitachi High-Tech Corp., Tokyo, Japan) was utilised to study the cross section, and thickness of the doped layer and the surface image was observed using an optical microscope and SEM. Energy dispersive X-ray (EDX, Oxford Instruments PLC, Oxford, UK) was used to obtain the elemental concentration in the sample. The crystallinity of the doped layer was investigated using a Philips X’Pert X-ray diffraction (XRD, Philips, Amsterdam, The Netherlands). A prism coupler was employed to measure the refractive index of the doped layer and verify the doped layer thickness. Photoluminescence (PL) emission spectra were recorded using Edinburgh Instruments FLS920 series spectrometer (Edinburgh Instruments Ltd., Livingston, UK) with a diode laser at a wavelength of 980 nm as an excitation source. Time-resolved PL spectra using laser source pulsed with a 100 ms period and a pulse width of 10 µs was also used to determine PL lifetime. Surface analysis, including elemental concentration measurement, was performed using X-ray photoelectron spectroscopy (Thermo Scientific K-alpha XPS, Waltham, MA, USA) with a monochromatic AlKα (1486.6 eV) source. Analysis and peak fitting were carried out using CasaXPS software (v.2.3.16). Renishaw inVia micro (Renishaw, New Mills, UK) with an excitation wavelength of 514 nm and a power of 25 mW was used to collect the Raman spectra.

## 3. Results and Discussion

### 3.1. Er-Doped Tellurite Modified Silica (EDTS) on SiO_2_-on-Silicon Substrate

Figure 1a displays a backscattered (BSE) cross-sectional SEM image of bare SOS substrate, while Figure 1b–d show BSE cross-sectional SEM images for samples fabricated when substrates were heated at temperatures of 400 °C (B400), 570 °C (B570), and 700 °C (B700). The thickness of the original silica layer (Table 1) remained at ~1 μm, proving that no doping process actually occurred for sample B400, and the layer obtained was only typical deposited film. This finding indicated that a temperature of 400 °C is insufficient for the incoming plasma plume to modify the strong covalent bonds of silica. The film formed on the SOS was only Er-TZN glass film, with compositions presented in Table 2, where the Te concentration was much higher than the Si concentration. The surface for the B400 sample appeared to be very rough with clusters of microparticles. Such film is unsuitable for EDWA because it could cause significant surface scattering and ultimately propagation loss for laser signals.

Figure 1c,d represent a cross section of samples produced at higher temperatures demonstrating the formation of Er-doped tellurite silica layer labelled as EDTS with thickness increasing with temperature. The elemental composition of the EDTS obtained from EDX-SEM and XPS for samples B570 and B700, as shown in Table 2, proved that the EDTS consisted of combinations of elements from the target material and silica from the substrate surface. This finding indicated that cations, such as Te, Zn, Na, and Er, were removed from the target material and diffused into silica, thereby modifying the original silica network during the ULPD process. However, Table 2 shows that the sample where the substrate was heated at 700 °C had a lower Te concentration than the sample prepared at 570 °C because of the volatility of Te that caused severe depletion through evaporation at a high temperature [22,23]. Nonetheless, the high temperature used to heat the substrate gave way to a relatively higher concentration of elements from the target glass, except Te, to dissolve into the silica.

As shown in Table 1, the thickness and refractive index of sample B400 could not be accurately determined by the prism coupler because of its rough surface. Meanwhile, the higher temperature used caused the materials from Er–TZN to penetrate more into the silica, thus causing a thicker EDTS. The reduced concentration of Te in sample B700, which is a heavy element in the EDTS, caused the lower refractive index of this sample than B570. Therefore, the concentration of Te in the EDTS is an essential factor as it could control the refractive index, which, in turn, could contribute to the design of integrated optical waveguides by using the ULPD approach presented in this work.

Figure 2 shows the XRD patterns of SOS substrate and SOS doped with Er-TZN when the substrate was heated to 400 °C, 570 °C, and 700 °C. For samples B570 and B700, the EDTS obtained were amorphous, except the peak at 2θ = 69°, corresponding to underlying crystalline silicon, Si (100) from SOS substrate. Meanwhile, for B400, the deposited film was clearly crystalline with a distinct peak at 2θ = 19.16°, 28.86°, 49.02°, and 59.68°. The peaks at 19.16° and 59.68° refer to Zn_2_Te_3_O_8_ based on the ICCD reference code: 04-012-2189 with each having miller indices of (111) and (332), whilst the peaks at 28.86° and 49.02° matched with Na_2_TeO_3_ with miller indices of (022) and (242), respectively (ICDD reference code: 00-035-1263).

The photoluminescence (PL) emission spectra of Er^3+^-ions in the samples were measured using a 980 nm diode laser as the excitation source. The PL emission spectra of the layers produced at various temperatures are represented in Figure 3. For sample B400, the spectrum shape obtained was clearly different from that of samples B570 and B700. The FWHM in sample B400 was broader (Table 3) than in samples B570 and B700 and almost similar to that reported for tellurite glasses [24,25,26]. For samples B570 and B700, the obtained FWHM was 20 nm and similar to the FWHM of other previously reported Er^3+^-doped silicate glasses, such as phosphosilicate, soda-lime silicate, and borosilicate [27,28]. This also confirms, in addition to elemental analysis, that the Er-TZN had permeated into the silica glass network and transformed it into silicate glass. Sample B400 had a lower PL lifetime (5.26 ms) than samples B570 (12.29 ms) and B700 (11.12 ms). The obtained PL lifetime for the Er-doped tellurite glass layer was obviously much lower than erbium-doped silicate glass as reported for other erbium-doped tellurite glasses, which are TeO_2_–GeO_2_–Na_2_O–ZnO–Er_2_O_3_ (5.7 ms) [29], TeO_2_–WO_3_–Na_2_O–Er_2_O_3_ (3.46 ms) [30], and TeO_2_–WO_3_–Na_2_O–Nb_2_O_5_–Er_2_O_3_ (3.7 ms) [31]. The lower PL lifetime for sample B400 is due to it being essentially a tellurite host material with stoichiometry similar to the target glass. PL lifetime is well known to also depend strongly on host material [32,33]. In particular, it is closely related to the refractive index of the host material based on the Judd–Ofelt theory. According to the theory, lifetime has an inverse relationship with refractive index [34,35,36]. Given that tellurite glass has a higher refractive index than silicate glass, it exhibited a lower lifetime, and this is the main reason that the PL lifetime for sample B400 was far lower than that for samples B570 and B700.

Figure 4 shows the XPS survey scan for samples fabricated using different substrate temperatures. For sample B400, the Te peak was obviously the highest among the samples, proving that tellurite is the predominant host material in the deposited layer. The absence of Si in sample B400 also confirms that there was no reaction in underlying SiO2 layer. For samples B570 and B700, the Er concentration was too low, and it could not be detected in the spectra survey. Er peaks that are often identified from Er 4d overlap with Si 2s peaks, which are more pronounced because of much higher silicon concentrations than the Er in samples B570 and B700. Furthermore, no peaks other than carbon (C 1s) and species from Er-TZN and silica were detected in the spectrum survey obtained. This finding indicated that no contaminants were present in the upper layer. For sample B700, the Te 3d_5/2_, and Te 3d_3/2_, peaks were very low compared with those in B570, and Te 2p_1/2_ and Te 2p_3/2_ peaks were absent, which validated the argument on the reduction of Te content in EDTS at higher temperatures.

Table 2 tabulates the element concentration for the surface layer of samples B400, B570, and B700, as calculated using a high-resolution scan of O 1s, Si 2p, Na 1s, Zn 2p_3/2_, Te 3d_5/2_, and Er 4p_3/2_. Even though the value for the element concentrations derived from XPS was slightly different from EDX-SEM, the trend was still quite similar. Figure 5 shows a high-resolution scan of Te 3d_5/2_ for a sample fabricated with varying substrate temperatures. As expected, the high-resolution scan of Te 3d_5/2_ for B400 was slightly different from that of the EDTS. No Te metal could be detected for this deposited layer, indicating that the amount of oxygen is sufficient in the layer. Surprisingly, TeO_3_ was dominant in this network. The peak located at ~574 eV is typically attributed to tellurium suboxide [37,38]. For the B700 sample, the low density of the tellurium species in EDTS made the scan results have much noise and made deconvolution processes difficult, possibly causing it to be less accurate.

Figure 6 shows the Raman spectra for samples prepared at different substrate temperatures. Samples B570 and B700 reported a wavenumber peak of 521 cm^−1^ attributed to single-crystalline silicon derived from the substrate [39,40]. This peak appeared to be lower for sample B700 because the EDTS for this sample was thicker than that for B570, which, in turn, caused the signal from the Si substrate to be weak. By contrast, the Si peak was invisible in sample B400 because of the thick layer of deposited Er-TZN and the SiO_2_ layer on a silicon substrate. Deconvolution of the spectrum was carried out for sample B570 to extract information about the hidden peaks for the EDTS layer.

The Raman spectra for sample B570 were deconvoluted into nine Gaussian bands (A–I), and the results are shown in Figure 7. The peak at energies lower than 250 cm^−1^ is named band A and considered a boson peak [41,42]. The Raman peak observed near band B (272 cm^−1^) is associated with a bending vibration of the TeO_3_ trigonal pyramid (tp) [43,44,45]. The appearance of TeO_3_ was also detected by XPS, as shown in Figure 5. Band C, which was in the range of 430–500 cm^−1^, is commonly associated with the Si-O-Si bending of Q_4_ species [39,46,47]. The peak around band E (590–650 cm^−1^) is attributed to the Si-O-Si bridges (bending vibration) between two Q_2_ species [46,48]. The emergence of a peak in the region of bands F (770–790 cm^−1^) and G (900–1000 cm^−1^) is often assigned as antisymmetric Si vibration in tetrahedral oxygen cage (Q_4_) [39,46,49] and Si-O-Si stretching of Q_2_ components, respectively [46,50]. The bands at a higher frequency located at bands H (~1160 cm^−1^) and I (~1544 cm^−1^) are associated with Er-related fluorescence, whilst the prominent peak at band D (521 cm^−1^) is referred to as silicon from the substrate. Meanwhile, the peak around ~250–280 cm^−1^ associated with TeO_3_ seemed more pronounced for sample B570 than for sample B700 (Figure 6) because of the higher Te density in B570.

### 3.2. Doped Layer on Si_3_N_4_-on-Silicon Substrate

Figure 8a shows a BSE cross-sectional SEM image of a Si_3_N_4_-on-silicon substrate prior to ULPD trials, whilst SEM images for samples K470, K520, K570, K600, and K650 are shown in Figure 8b–f. As Figure 8b illustrates, deposition of Er-TZN onto the substrate occurred. The layer formed on the Si_3_N_4_ substrate appeared uniform and smooth. However, only a small amount of Er-TZN successfully penetrated Si_3_N_4_, indicating that a temperature of 470 °C is not sufficiently high to break the Si-N bond. Only a few of the energetic elements of Er-TZN managed to enter into the Si_3_N_4_ molecular network, as could be seen around the materials interface. According to the SEM images in Figure 8c–f, for samples fabricated with a higher temperature of 520–650 °C, Er-TZN successfully diffused into the Si_3_N_4_. However, all the images of the doped layer demonstrated that the doping was not as uniform as doping silica with Er-TZN. Although elements from the target material managed to penetrate Si_3_N_4_, they were discretely distributed, as depicted by the grayscale variations in the doped layers. This largely inhomogeneous layer appeared to be partly porous, and, in certain areas, an accumulation of specific elements was observed. The SEM images of the surface of the doped layer also showed that it was very uneven and rough, as shown in Figure 9.

The thicknesses of the upper layers for all of the samples are summarised in Table 4. For sample K470, the thickness of the Si_3_N_4_ underneath the upper layer was less than the original thickness of 1 μm, which indicated that doping occurred, even though only a small amount was found on the Si_3_N_4_ surface. For samples K520, K570, K600, and K650, the upper layer thicknesses could not be precisely determined, and they showed large errors due to their very rough surfaces. However, the estimated thickness obtained showed a linear relationship between thickness and the temperature used to heat the substrate. Additionally, the thickness of the Si_3_N_4_ underneath the doped layer became smaller with an increase in substrate temperature. This finding indicated that the material target could react with Si_3_N_4_ more thoroughly at higher temperatures. The refractive index and thickness, which were supposedly measured using a prism coupler, could not be measured because of the uneven surface of the doped layer.

Figure 10 shows two different positions in sample K470 measured by EDX-SEM to obtain the elemental concentrations at these particular positions, and the results are shown in Table 5. As the upper layer of sample K470 was only a deposited layer, it was expected to have the properties of tellurite-based glass used as a target material. The high concentration of Te at position 1 confirmed this expectation. For position 2, which is located near the interface of the deposited region and Si_3_N_4_ but within the Si_3_N_4_, some elements from the target material were mostly oxygen. The smaller atomic size of oxygen than that of Te, Zn, Na, and Er enabled a few energetic ions/atoms of oxygen to successfully penetrate the Si_3_N_4_ and oxidise some of it. However, the lower temperatures evidently hindered deeper modification of the Si_3_N_4_ layer further. The area scan performed for the sample to determine the distribution of elements in sample K470 is shown in Figure 11. The negligible presence of elements Si and N in the upper layer confirmed that the upper layer consisted of elements from Er-TZN. Besides, the area scan obtained showed an intermediate region between the deposited layer and Si_3_N_4_ composed of Si, O, Te, Zn, Na, Er, and N. However, the area scan for Er was less obvious because of its deficient concentration, resulting in considerable noise in the scan results.

An area scan was performed to view the entire elemental distribution present in K600 (Figure 12). The elements were clearly distributed unevenly. Surprisingly, the concentration of N in the doped layer was very low compared with its concentration in Si_3_N_4_, which was supposedly around ~57.1 %. Nitrogen was considered lost in the form of nitrogen gas when the tellurite glass reacted with Si_3_N_4,_ and this phenomenon was also reported by Watanabe et al. [51]. Failure to obtain a very uniform doped layer, such as in the case of EDTS, was probably due to the substrate temperature not being sufficiently high. The melting temperature of Si_3_N_4_ (1900 °C) was higher than that of silica (1710 °C). Therefore, higher temperatures may be required to allow the dissolution of Er-TZN into Si_3_N_4_ with homogeneous layer formation. Additionally, the failure to obtain homogeneously doped layers may be due to the SiO_2_ amorphous network, which is somewhat different from Si_3_N_4_. Unlike the local structure of silica, which contains adjustable and flexible Si-O-Si- bridging bonds, Si_3_N_4_, consists of Si-N-Si bonds that are rendered rigid as N requires bonding with three Si rather than two to form a stable configuration. As a consequence, its network structure is much more constrained than that of silica [52], making it difficult for the elements from Er-TZN to diffuse into Si_3_N_4_ and modify it. This leads to the formation of pores and cavities and the accumulation of certain elements for Si_3_N_4_ samples. A higher temperature could be expected to help loosen the strong Si-N bonds. Nonetheless, the more rigid Si_3_N_4_ structure caused it to have higher internal stress levels and thus to crack more easily. This problem was not observed for samples fabricated using an SOS substrate. Figure 13a,b show the surface images of samples that used different substrates (SiO_2_ and Si_3_N_4_) with the same process parameters and target material. The images were taken under an optical microscope near the edge of the doped layer. The findings clearly showed that the undoped Si_3_N_4_ layer on silicon cracked after the sample fabrication process.

Figure 14 shows the XRD patterns of the Si_3_N_4_-on-silicon substrate and samples K470, K520, K570, K600, and K650. The patterns showed that all samples were in a mixed amorphous–crystalline phase. The indexed peaks assigned to the possible crystalline structures are also shown in Figure 14. The 2θ peak located at approximately ~69° is a crystalline Si (100) peak originating from the silicon substrate. For sample K470, eleven peaks were detected, namely, 23.38°, 27.87°, 38.63°, 40.61°, 43.61°, 46.42°, 48.73°,49.95°, 57.24°, 63.11°, and 65.98°, which correspond to Te (ICCD reference code: 00-036-1452). The appearance of Te crystallite peaks is attributable to the transition layer between the deposited and Si_3_N_4_ layers. For the doped layer of samples K520–K650, the intensity and number of crystalline peaks increased when higher substrate temperatures were used because crystallisation occurs more easily at high temperatures [53,54]. The crystalline peaks for SiO_2_ (ICCD reference code: 00-039-1425) and Na_2_Zn_3_ (SiO_4_)_2_ (ICCD reference code: 00-012-3700) started to appear when the sample was heated at 600 °C, and they became more pronounced at higher temperatures. In addition, for the doped layer, the crystalline phase of Te increased in number and intensity with respect to increasing temperature.

Figure 15 presents the Raman spectra for the Si_3_N_4_-on-silicon substrate and samples fabricated with various substrate temperatures. The Raman spectra for sample K470 (Figure 15b), comprising tellurite-based glass, were different compared with samples prepared at higher temperatures. As shown in Figure 15c–f, all Raman spectra showed metallic tellurium peaks (122 and 141 cm^−1^). This finding indicated that the tellurium cluster was likely to form within this doped layer and coincides with the findings from XRD. However, these two peaks were less pronounced in sample K470, suggesting that most of the Te in this sample was present in oxide form rather than metallic form. The stoichiometry of Er-TZN was better preserved in the deposited film than in the doped layer. The Raman spectrum for K470 was similar to that for B400 (Figure 6a) because both of them were deposited tellurite-based layers. The decrease in Si peak intensity from samples K520 to K650 indicated that the thicknesses of the doped layer increased for samples fabricated at higher substrate temperatures. The Raman spectra for samples K520, K570, K600, and K650 (Figure 15c–f) were similar to the EDTS, and only the appearance of peaks at 122 and 141 cm^−1^ distinguished them. Therefore, the assignment of all other peaks in the spectra was the same as discussed for the EDTS, as the contribution of N was not detectable.

Figure 16 displays the XPS survey spectrum for sample K570. The spectrum pattern obtained resembled that of the EDTS. The absence of N confirmed that N is released when Er-TZN reacts with Si_3_N_4_. The distinct peak of N 1s that was expected at a binding energy of around 398 eV was not observed in the obtained spectra. Therefore, N is present only in very small quantities or virtually absent in the doped layer. The presence of Si in the spectral survey verified once again that Er-TZN successfully entered into the Si_3_N_4_ layer and formed a predominantly silicate-based layer.

The PL emission spectra for the samples are shown in Figure 17. For K470, the FWHM of the spectra was broader than that of the others, also indicating that the upper layer was a tellurite-based material. The low PL lifetime (4.97 ms) and an FWHM value of approximately 33 nm (Table 6) obtained were comparable with those reported for tellurite glass [24,55]. As shown in Table 6, samples K520, K570, K600, and K650 had a PL lifetime of between 9 and 10 ms and FWHMs of 20 nm. This finding indicated that the doped layer was silicate based, as with the EDTS. If the doped layer is Si_3_N_4_ based, it could show a lower lifetime in the range of 0.2–7 ms with an FWHM broader than that of a typical Er-doped silicate-based material [56,57,58]. As shown in Table 6, the higher PL intensity and the lower lifetime of samples prepared at higher substrate temperatures signified that the Er density is higher in the doped layer when the substrate is heated to a higher temperature.

## 4. Conclusions

In summary, the ULPD technique successfully integrated two immiscible materials (Er-TZN and silica) that resulted in a homogeneous layer of the mixture, referred to as EDTS. However, the quality of the resulting film and the effectiveness of the doping process that occurred were highly dependent on the substrate temperature. The temperature of 400 °C was not sufficient for the incoming plasma plume to modify the strong covalent bonds of silica, and the doping process could not take place. The higher temperature used caused the materials from Er-doped tellurite glass to penetrate more (except Te with smaller concentration) into silica, thus creating a thicker film. The reduced concentration of Te in the doped silica on silicon (SOS) produced at 700 °C caused the refractive index to decline. However, Er-TZN doping into Si_3_N_4_ failed to show a homogeneous layer because Si_3_N_4_ has a high melting temperature and rigid structure compared with SiO_2_. A higher temperature could be expected to help loosen the strong Si-N bonds, but the rigid Si_3_N_4_ structure caused it to have higher internal stress levels and thus crack more easily.

## Figures and Tables

**Figure 1 nanomaterials-12-00919-f001:**
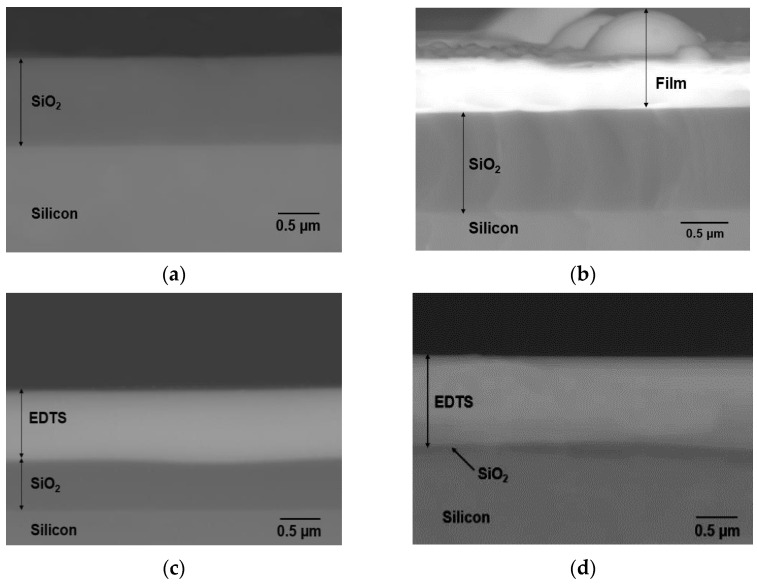
Backscattered cross-sectional SEM image of (**a**) bare SOS substrate and samples doped with Er-TZN when the substrate was heated at (**b**) 400 °C (B400), (**c**) 570 °C (B570), and (**d**) 700 °C (B700).

**Figure 2 nanomaterials-12-00919-f002:**
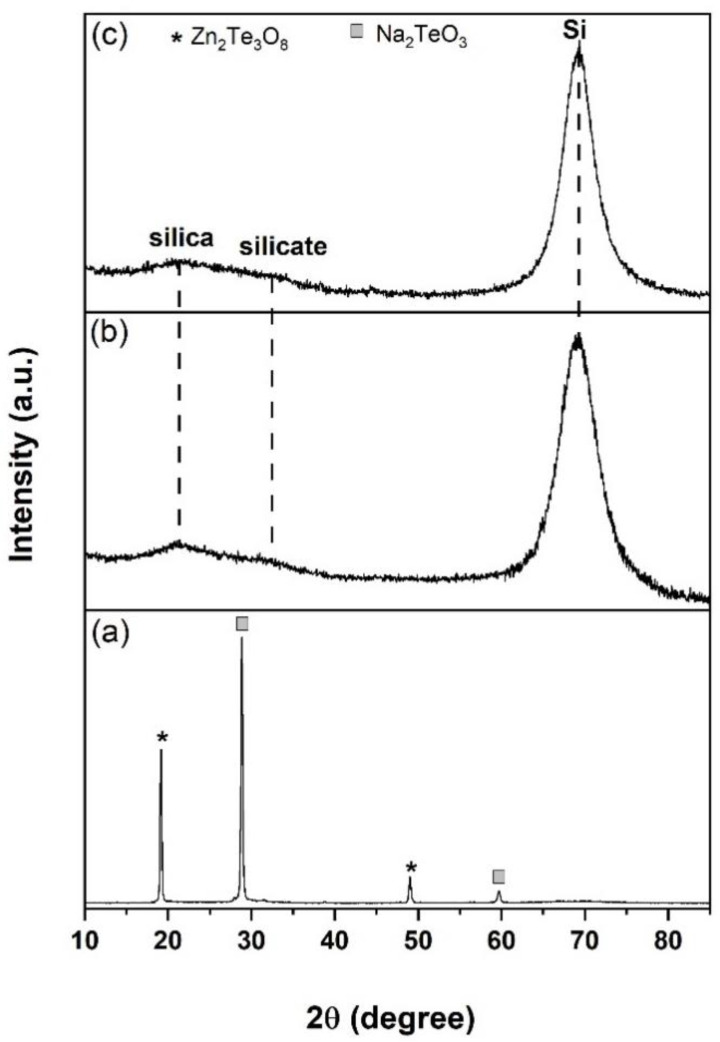
XRD patterns for samples prepared using temperatures of (**a**) 400 °C (B400), (**b**) 570 °C (B570), and (**c**) 700 °C (B700).

**Figure 3 nanomaterials-12-00919-f003:**
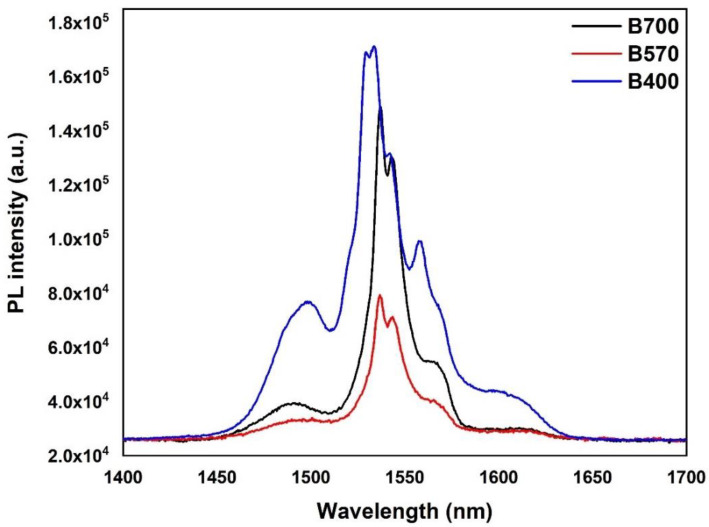
PL spectra of upper layer prepared at different substrate temperatures.

**Figure 4 nanomaterials-12-00919-f004:**
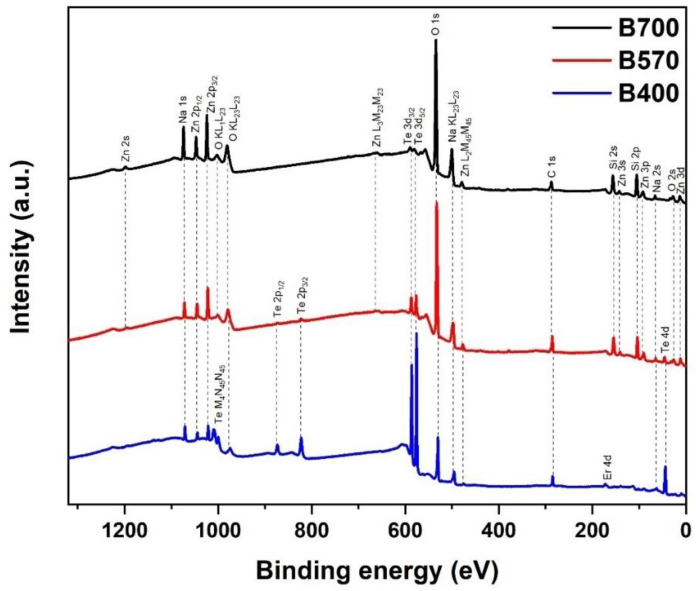
XPS survey scan for samples B400, B570 and B700.

**Figure 5 nanomaterials-12-00919-f005:**
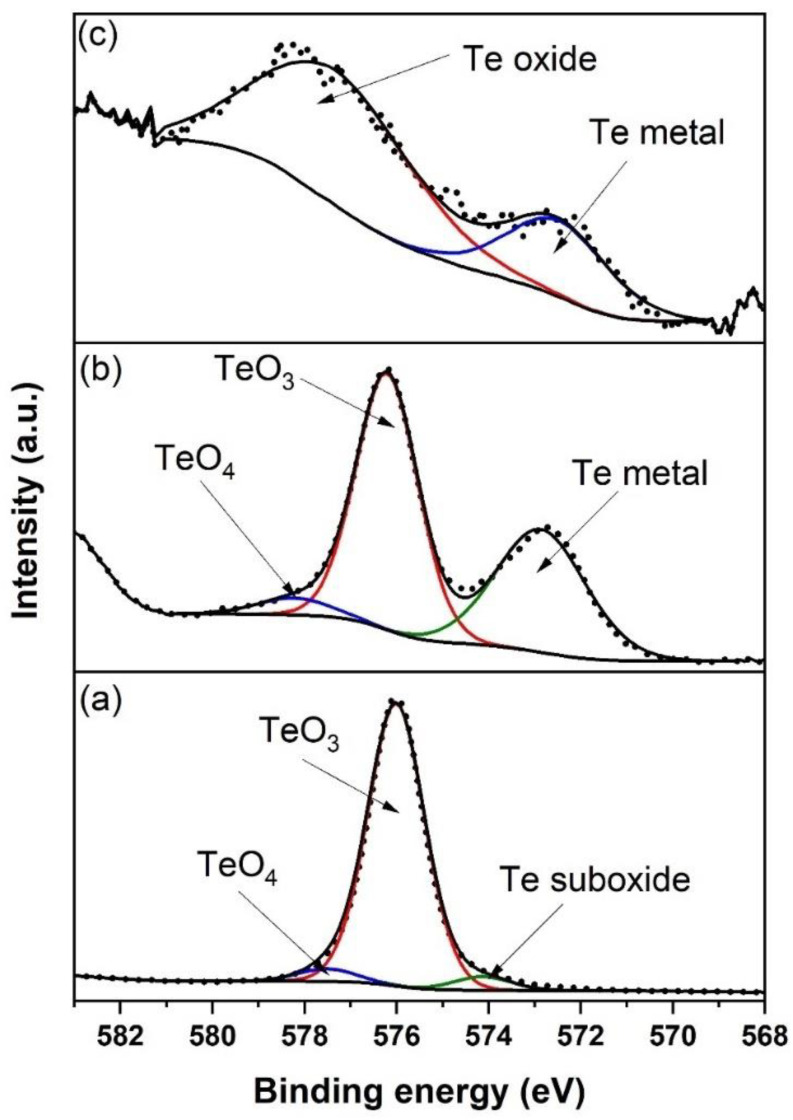
Te 3d_5/2_ spectra of the surface of EDTS layer for samples prepared at (**a**) 400 °C (B400), (**b**) 570 °C (B570), and (**c**) 700 °C (B700).

**Figure 6 nanomaterials-12-00919-f006:**
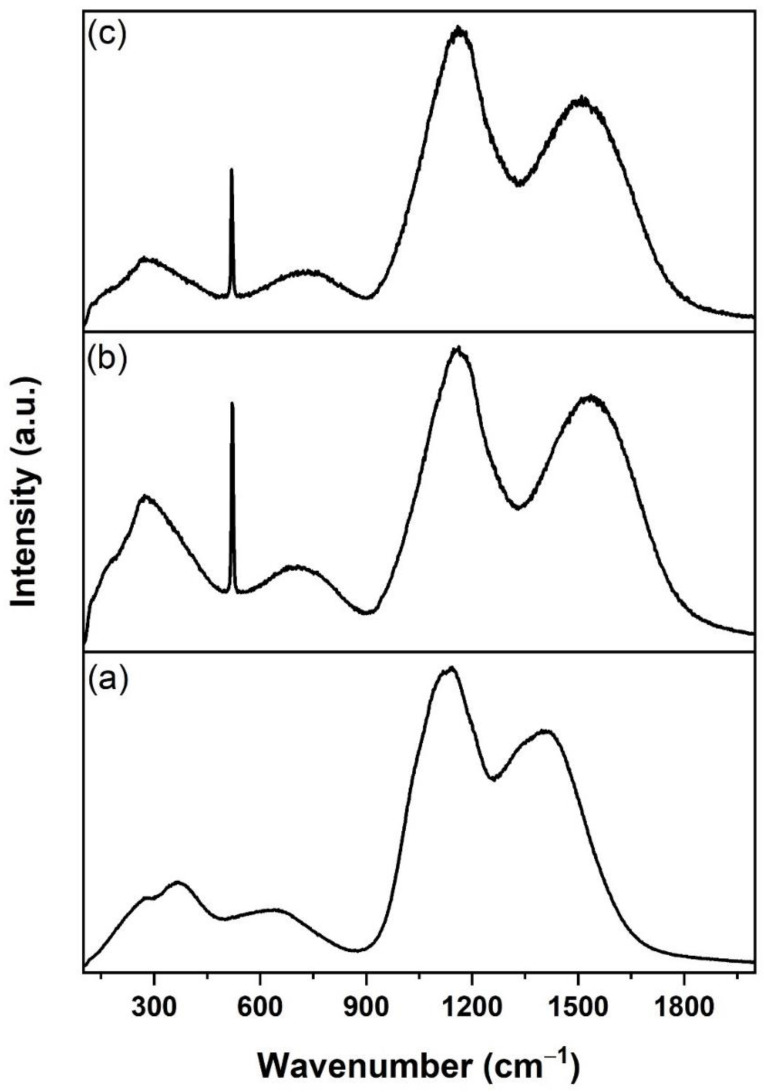
Raman spectra of samples (**a**) B400, (**b**) B570, and (**c**) B700. B570 and B700 samples show similar spectra.

**Figure 7 nanomaterials-12-00919-f007:**
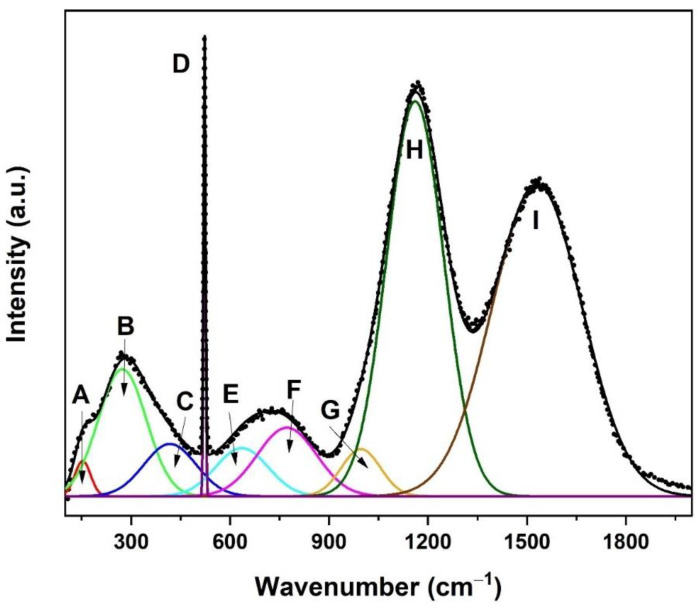
Deconvolution of Raman spectra of sample B570 in various Gaussian bands. The bands investigated in this work labelled as A–I are marked.

**Figure 8 nanomaterials-12-00919-f008:**
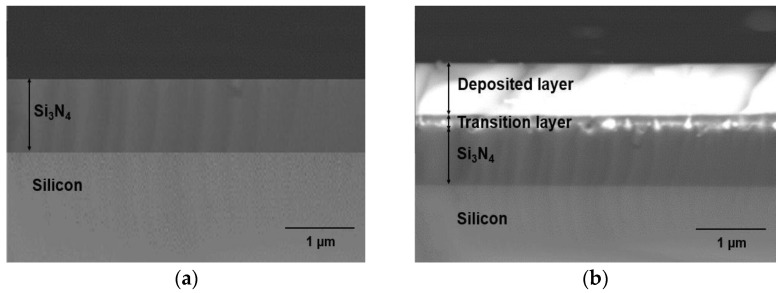
BSE cross-sectional SEM images of (**a**) bare Si_3_N_4_-on-silicon substrate, and samples doped with Er-TZN into Si_3_N_4_-on-silicon when substrate was heated at (**b**) 470 °C (K470) (**c**) 520 °C (K520), (**d**) 570 °C (K570), (**e**) 600 °C (K600), and (**f**) 650 °C (K650).

**Figure 9 nanomaterials-12-00919-f009:**
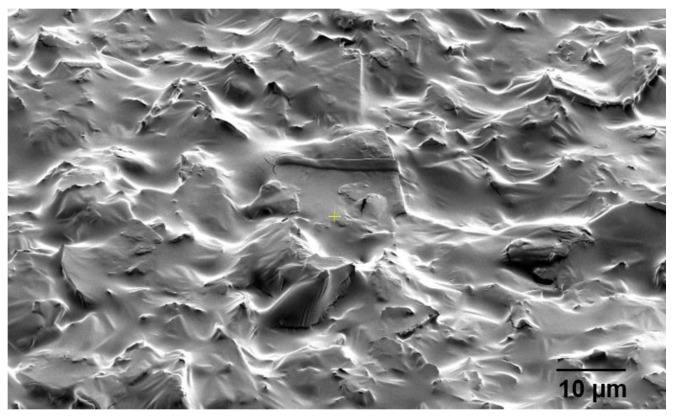
SEM surface morphology of the top surface of the doped layer for sample K600.

**Figure 10 nanomaterials-12-00919-f010:**
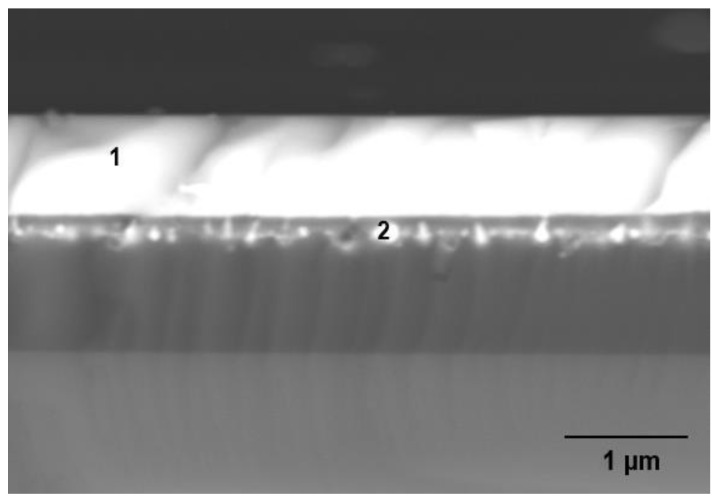
EDX-SEM measurement at two different locations for sample K470. Position 1 is the location where Te is abundant while position 2 is the location near the interface of the deposited region and Si_3_N_4_.

**Figure 11 nanomaterials-12-00919-f011:**
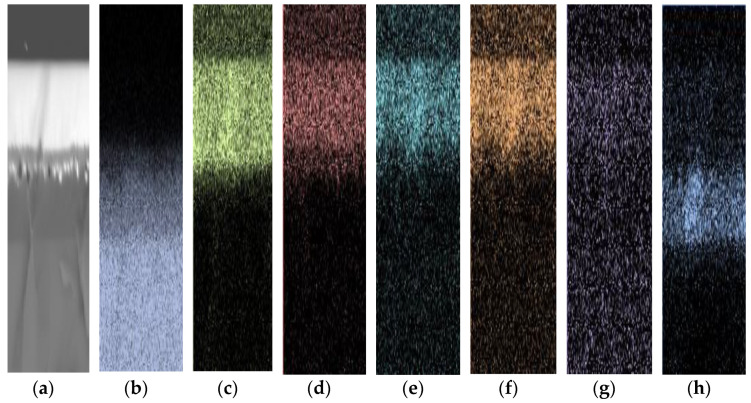
Area scan measured by EDX-SEM for sample K470 with (**a**) the area being measured. The distribution of species presented is that of (**b**) silicon, (**c**) oxygen, (**d**) tellurium, (**e**) sodium, (**f**) zinc, (**g**) erbium, and (**h**) nitrogen.

**Figure 12 nanomaterials-12-00919-f012:**
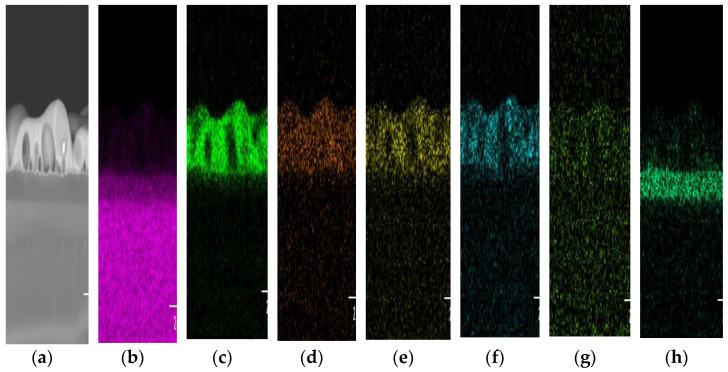
Area scan measured by EDX-SEM for sample K600 with (**a**) the area being measured. The distribution of species presented is that of (**b**) silicon, (**c**) oxygen, (**d**) tellurium, (**e**) sodium, (**f**) zinc, (**g**) erbium and (**h**) nitrogen.

**Figure 13 nanomaterials-12-00919-f013:**
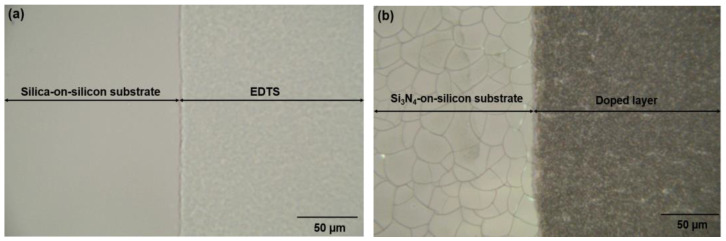
Surface images taken by optical microscope for samples fabricated with the same process parameters and target material for (**a**) silica-on-silicon (**b**) Si_3_N_4_-on-silicon substrate.

**Figure 14 nanomaterials-12-00919-f014:**
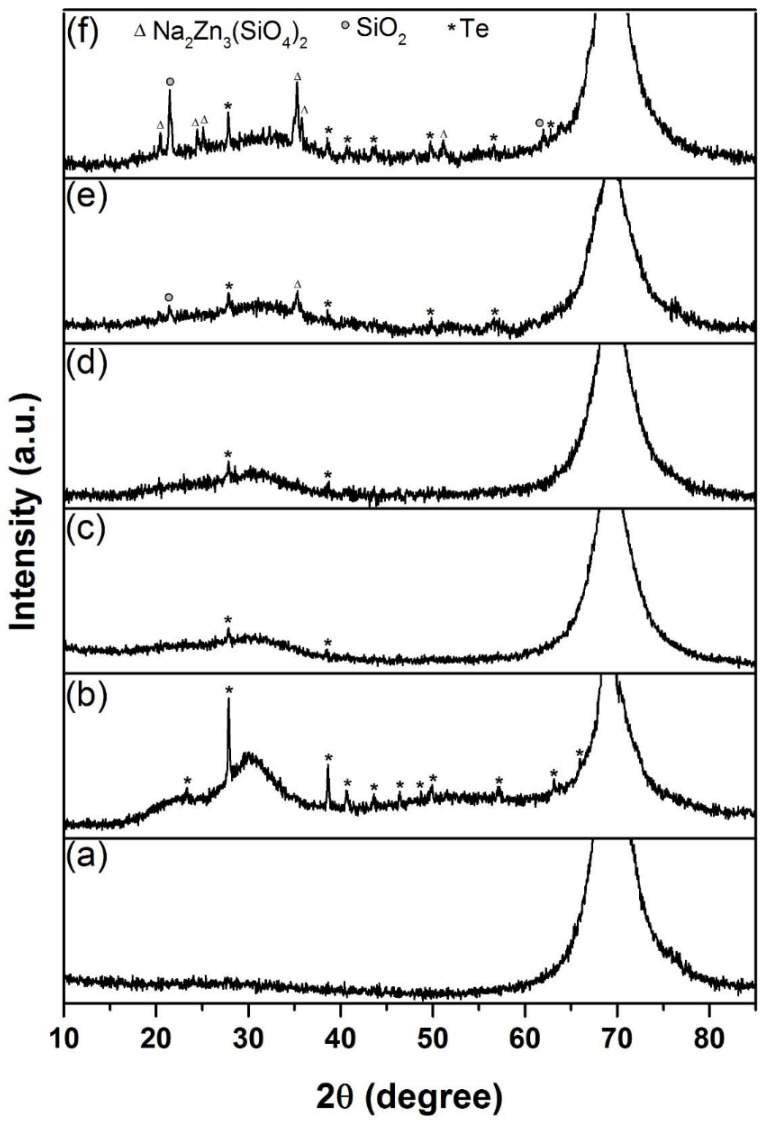
XRD patterns of (**a**) Si_3_N_4_-on-silicon substrate and samples (**b**) K470, (**c**) K520, (**d**) K570, (**e**) K600 and (**f**) K650.

**Figure 15 nanomaterials-12-00919-f015:**
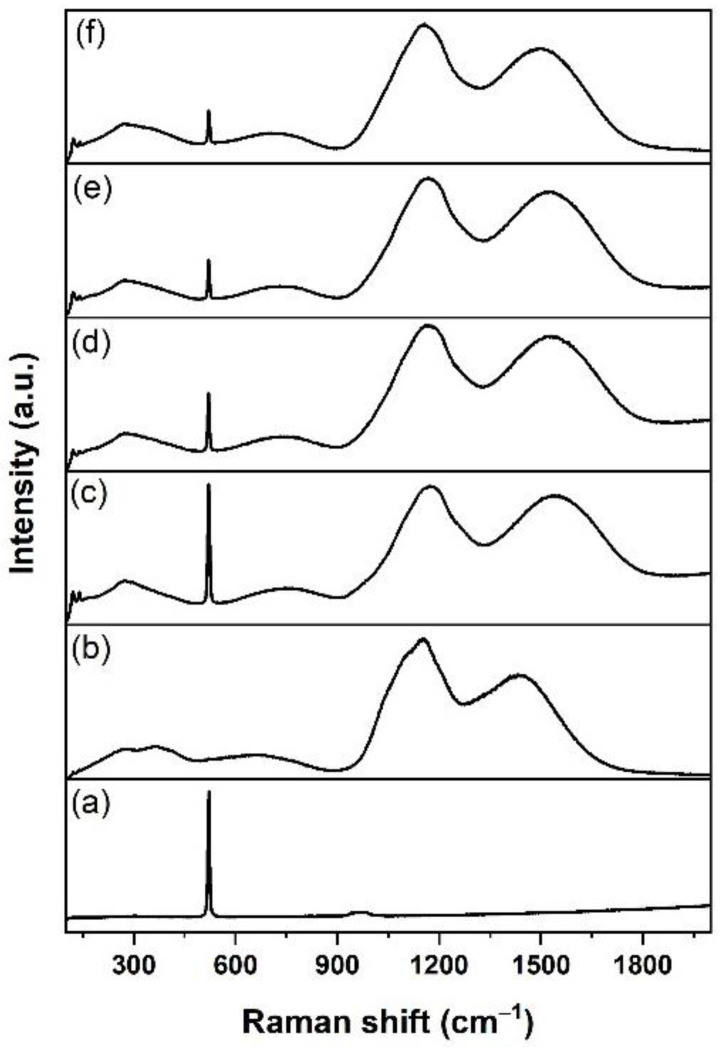
Raman spectra of (**a**) Si_3_N_4_-on-silicon substrate and samples (**b**) K470, (**c**) K520, (**d**) K570, (**e**) K600 and (**f**) K650.

**Figure 16 nanomaterials-12-00919-f016:**
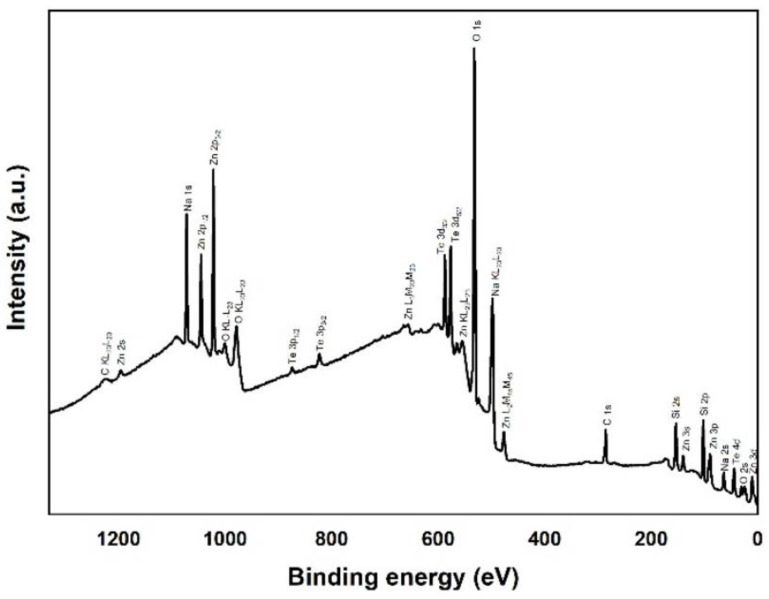
XPS survey spectrum for the sample prepared using a substrate temperature of 570 °C (K570).

**Figure 17 nanomaterials-12-00919-f017:**
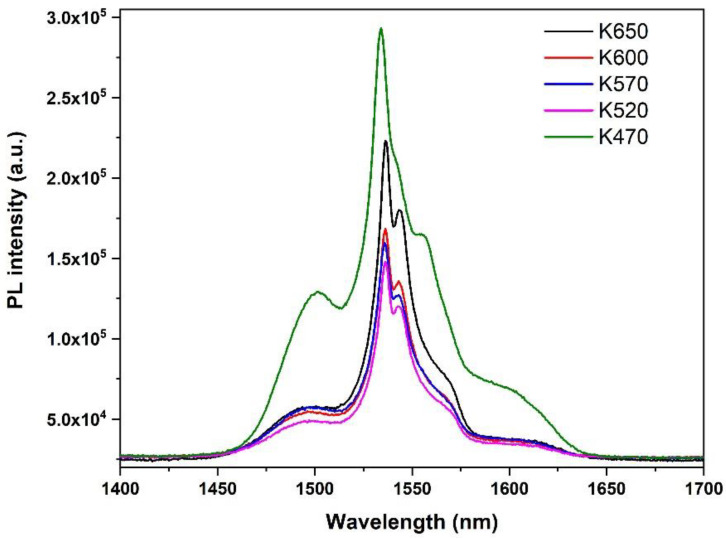
Room-temperature PL spectra of samples K470, K520. K570, K600 and K650. The broad shape for sample K470 indicates it has a different host from others.

**Table 1 nanomaterials-12-00919-t001:** Variation of the upper layer and remaining silica layer thickness and refractive index prepared at different substrate temperatures.

Sample	EDTS Thickness Measured with SEM (µm)	EDTS Thickness Measured with Prism Coupler (µm)	SiO_2_ Thickness of the Layer below EDTS Measured with SEM (µm)	Refractive Index
B400	-	-	1.03 ± 0.03	-
B570	0.81 ± 0.06	0.83 ± 0.05	0.56 ± 0.06	1.5587 ± 0.0004
B700	1.10 ± 0.03	1.10 ± 0.01	0.22 ± 0.06	1.5285 ± 0.0002

**Table 2 nanomaterials-12-00919-t002:** Percentage of components existing in the upper layer of SOS substrate for samples fabricated using different substrate temperatures obtained from EDX-SEM and XPS.

Element	Elemental Concentration (at. %)					
	B400 (at. %)		B570 (at. %)		B700 (at. %)	
	EDX-SEM	XPS	EDX-SEM	XPS	EDX-SEM	XPS
O	50.69	52.24	61.01	58.51	60.91	58.40
Si	1.67	1.85	21.47	28.90	20.99	28.79
Na	7.94	10.18	8.57	5.72	9.68	7.08
Zn	6.60	6.95	6.77	3.50	7.14	4.57
Te	32.21	27.08	1.74	2.89	0.73	0.59
Er	0.89	1.70	0.44	0.48	0.55	0.57

**Table 3 nanomaterials-12-00919-t003:** FWHM and PL lifetime for film fabricated at different substrate temperatures.

Sample	FWHM (nm)	PL lifetime (µm)
B400	38	5.26
B570	20	12.29
B700	20	11.12

**Table 4 nanomaterials-12-00919-t004:** The thickness of the upper layer and Si_3_N_4_ underneath, as measured by SEM for samples K470, K520, K570, K600, and K650.

Sample	Thickness (µm)	
	Upper Layer	Si_3_N_4_ under Upper Layer
K470	1.1 ± 0.1	0.87 ± 0.04
K520	1.5 ± 0.4	0.74 ± 0.03
K570	1.5 ± 0.3	0.66 ± 0.03
K600	1.8 ± 0.3	0.60 ± 0.05
K650	2.0 ± 0.3	0.57 ± 0.09

**Table 5 nanomaterials-12-00919-t005:** The elemental concentration of sample K470 at two locations, as shown in Figure 10.

Sample	Concentration (µm)	
	Position 1	Position 2
O	58.79	15.86
Si	2.25	39.49
Te	23.70	2.83
Zn	5.78	1.23
Na	8.45	1.54
Er	0.65	0.02
N	0.38	39.03

**Table 6 nanomaterials-12-00919-t006:** PL lifetime and FWHM of samples K470, K520, K570, K600 and K650.

Sample	FWHM (nm)	PL Lifetime (µm)
K470	33	4.97
K520	20	10.43
K570	20	9.94
K600	20	9.71
K650	20	9.59

## Data Availability

Data are contained within the article.

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
