# Peer review of "Effect of Substrate Temperature on Morphological, Structural, and Optical Properties of Doped Layer on SiO2-on-Silicon and Si3N4-on-Silicon Substrate"

_nanomaterials, 2022, doi:10.3390/nano12060919_

Round 1

Reviewer 1 Report

The purpose of this study is to explore a miniaturized broadband amplification device that can replace EDFA amplifier. For this, the preparation of thin films deposited on two substrate materials is studied. The properties of thin films deposited on substrates at different temperatures are compared and analyzed. Through different tests, the relatively ideal substrate temperature is obtained. This research has certain reference significance for the preparation of thin films by UPLD technology. However, from the current research results, it is difficult to see the broadband amplification characteristics of the prepared films, and there is still a large gap with the existing widely used EDFA. Therefore, starting from the realization of the target application mentioned by the author, I hope the paper can increase the analysis of the impact on the performance of broadband amplification. Moreover, I think the doping concentration obtained by the author is not high, so it is difficult to achieve high gain performance of short cavity. Is it difficult to achieve the research goal mentioned in the paper. In any case, I think the author should add some broadband amplification test results.

Author Response

1) Extensive editing of English language and style required

    Proofreading for the English language has been done (extensive editing).

2) The purpose of this study is to explore a miniaturized broadband amplification device that can replace EDFA amplifier. For this, the preparation of thin films deposited on two substrate materials is studied. The properties of thin films deposited on substrates at different temperatures are compared and analyzed. Through different tests, the relatively ideal substrate temperature is obtained. This research has certain reference significance for the preparation of thin films by UPLD technology. However, from the current research results, it is difficult to see the broadband amplification characteristics of the prepared films, and there is still a large gap with the existing widely used EDFA. Therefore, starting from the realization of the target application mentioned by the author, I hope the paper can increase the analysis of the impact on the performance of broadband amplification. Moreover, I think the doping concentration obtained by the author is not high, so it is difficult to achieve high gain performance of short cavity. Is it difficult to achieve the research goal mentioned in the paper. In any case, I think the author should add some broadband amplification test results.

The main purpose of the paper is to study the effect of the substrate temperature of the doped layer on SiO2-on-silicon and Si3N4-on-silicon substrate. This study focuses more on the blending of femtosecond (fs) laser-produced plasma from a TeO2 (target) based glass with a SiO2 and Si3N4 (substrate) to form a silicate glass layer and how the substrate temperature can affect the structural, morphological, and optical properties of the resulting doped layer. The obtained doped layer has a great potential to be used in EDFA/EDWA applications. We still do not have results for the broadband amplification test but will be set up in the future.

Reviewer 2 Report

The article is devoted to the study of the physicochemical properties of telluride substrates doped with Er3+. As a doping method, the method of ultrafast laser plasma, as well as its effect on the substrate surface, was used. These studies are of scientific interest and practical significance, since the selected compositions have a high potential for industrial use. However, before the article is accepted for publication, the authors should make changes and answer the questions posed.

1. In the introduction, the authors should write in more detail the purpose and relevance of this work, taking into account the latest achievements in this field.
2. When conducting experiments, the authors should take into account the melting temperatures of the substrates in order to control the doping process. In particular, the presented results at a temperature of 400 C are not clear, since it is known that alloying processes are difficult at these temperatures, due to small changes in the substrate surface.
3. The authors should explain what is the reason for the formation of pores and cavities for Si3N4 samples?
4. The authors should provide more convincing data on doping; the presented data of energy-dispersive analysis raise doubts about the correct interpretation of the presence of alloyed elements.
5. X-ray data require clarification whether the presence of alloyed elements was observed using this method.

Author Response

Point 1: Extensive editing of English language and style required

Response 1: Proofreading for the English language has been done (extensive editing).

Point 2: In the introduction, the authors should write in more detail the purpose and relevance of this work, taking into account the latest achievements in this field.

Response 2: Previous studies have proven Er-TZN to be successfully doped into silica [13], [15], [16], [18]–[20]. Since this process is also assisted by substrate heating during the doping process, selecting the appropriate temperature is very critical. Substrate temperature is a significant parameter in ULPD because it activates the mobility of the ablated species for it to diffuse into the substrate and modify the host SiO2 and Si3N4 networks. Strong covalent bonds can be modified by the incoming plasma plume when appropriate temperatures are used to heat the substrate. Therefore, the effect of substrate temperature for these two substrates is worth studying as this effect in the ULPD technique has not yet been reported. The use of different substrate temperatures thus results in varying doped layer characteristics.

[13] Kamil,S.A.; Chandrappan, J.; Murray,M.; Steenson, P.; Krauss, T.F.; Jose, G. Ultrafast laser plasma doping of Er3+ ions in silica-on-silicon for optical waveguiding applications, Opt. Lett. 2016, 41,4684-4687.

[15] Chandrappan,J.; Murray,M.; Kakkar, T.; Petrik, P.; Agocs,E.; Zolnai, Z.; Steenson, D.P.; Jha,A.; Jose,G. Target dependent femtosecond laser plasma implantation dynamics in enabling silica for high density erbium doping. Sci. Rep. 2015,5, 14037.

[16] Chandrappan, J.; Murray, M.; Pe trik, P.; Agocs, E.; Zolnai, Z.; Tempez, A.; Legendre, S.; Steenson, D.P.; Jha,A.; Jose,G. Doping silica beyond limits with laser plasma for active photonic materials. Opt. Mater. Express 2015, 5, 2849-2861.

[18] Kamil,S.A.; Chandrappan, J.; Portoles, J.; Steenson, D.P.; Jose,G. Local structural analysis of erbium-doped tellurite modified silica glass with X-ray photoelectron spectroscopy. Mater. Res. Express 2019,6,086220.

[19] G. Jose, J. Chandrappan, S.A. Kamil, M. Murray, Z. Zolnai, P. Petrik, P. Steenson, T.  Krauss, Ultrafast laser plasma assisted rare-earth doping for silicon photonics, in: Conf. Lasers Electro-Optics, Optical Society of America, San Jose, California, 2016.

[20] J. Chandrappan, V. Khetan, M. Ward, M. Murray, G. Jose, Devitrification of ultrafast laser plasma produced metastable glass layer, Scr. Mater. 131 (2017) 37–41.

Point 3: When conducting experiments, the authors should take into account the melting temperatures of the substrates in order to control the doping process. In particular, the presented results at a temperature of 400 °C are not clear, since it is known that alloying processes are difficult at these temperatures, due to small changes in the substrate surface.

Response 3: The melting temperature of Si3N4 and silica is 1900 °C and 1710 °C, respectively (page 12, paragraph 1, line 338-339). The resulting plasma plume is believed to have a very high temperature that causes it to penetrate into the substrate even though the substrate temperature used is much lower than substrate melting temperature (not written in the manuscript).

The highest temperature that can be supplied by the equipment is 750 °C. We tried to use the lowest temperature to avoid evaporation of Te (not written in the manuscript).

Point 4: The authors should explain what is the reason for the formation of pores and cavities for Si3N4 samples?

Response 4: The melting temperature of Si3N4 (1900 °C) was higher than that of silica (1710 °C). Therefore, higher temperatures may be required to allow the dissolution of Er-TZN into Si3N4 with homogeneous layer formation. Additionally, the failure to obtain homogeneously doped layers may be due to the SiO2 amorphous network, which is somewhat different to that of Si3N4. Unlike the local structure of silica, which contains adjustable and flexible -Si-O-Si- bridging bonds, Si3N4, consists of- Si-N-Si- bonds that are rendered rigid as N requires bonding with three Si rather than two to form a stable configuration. As a consequence, its network structure is much more constrained than that of silica [52] making it difficult for the elements from Er-TZN to diffuse into Si3N4 and modify it. This leads to the formation of pores and cavities and accumulation of certain elements for Si3N4 samples (page 12, paragraph 1, line 338-348).

Point 5: The authors should provide more convincing data on doping; the presented data of energy-dispersive analysis raise doubts about the correct interpretation of the presence of alloyed elements.

Response 5: XPS measurements have been done to prove the presence of alloyed elements (Table 2, Figure 4 and Figure 16). The element concentration in Table 2 were calculated using a high-resolution scan of O 1s, Si 2p, Na 1s, Zn 2p3/2, Te 3d5/2 and Er 4p3/2.

Point 6: X-ray data require clarification whether the presence of alloyed elements was observed using this method.

Response 6: XPS measurements have been done to clarify the presence of alloyed elements (Table 2, Figure 4 and Figure 16).  EDX results also showed the layer consist of a mixture of element from target materials and substrate (SiO2 and Si3N4). Furthermore, Raman and PL measurement exhibit the silica-based spectra which prove the layer is a combination of Er-TZN and substrate. 

Round 2

Reviewer 2 Report

The authors answered all the questions, the article can be published.